# Psychotropic medications induced parkinsonism and akathisia in people attending follow-up treatment at Jimma Medical Center, Psychiatry Clinic

**Assefa Kumsa, Liyew Agenagnew, Beza Alemu, Shimelis Girma**  *

Department of Psychiatry, Institute of Health, Jimma University, Jimma, Ethiopia

* shimelisgirma@gmail.com

**Data Availability Statement:** The data generated for this study are included in this article and Dryad at https://doi.org/10.5061/dryad.1vhhmgqqb.

## Abstract

### Objective

To determine the magnitude and factors associated with psychotropic drug-induced parkinsonism and akathisia among mentally ill patients.

### Methods

A hospital-based cross-sectional study was conducted with a total of 410 participants attending a follow-up treatment service at Jimma Medical Center, a psychiatry clinic from April to June 2019. Participants were recruited using a systematic random sampling method. Drug-induced parkinsonism and akathisia were assessed using the Extra-pyramidal Symptom Rating Scale. Substance use was assessed using the World Health Organization Alcohol, Smoking, and Substance Involvement Screening Test. Data entry was done using EpiData version 3.1, and analysis done by the Statistical Package for Social Sciences version 22. Statistically, the significant association was declared by adjusted odds ratio, 95% confidence interval, and *p-value* less than or equal to 0.05.

### Results

The mean age of the respondents was 33.3 years (SD ± 8.55). Most of the participants 223 (54.4%) had a diagnosis of schizophrenia. The prevalence of drug-induced parkinsonism was 14.4% (95% CI: 11.0 to 18.0) and it was 12.4% (95% CI: 9.3 to 15.4) for drug-induced akathisia. The result of the final model found out drug-induced parkinsonism was significantly associated with female sex, age, type of antipsychotics, physical illness, and anti-cholinergic medication use. Similarly, female sex, chlorpromazine equivalent doses of 200 to 600 mg, combined treatment of sodium valproate with antipsychotic, and severe *khat/Catha edulis* use risk level was significantly associated with akathisia.

### Conclusion

One of seven patients developed drug-induced parkinsonism and akathisia. Careful patient assessment for drug-induced movement disorders, selection of drugs with minimal side

**Funding:** Financial expense for data collection was covered by Jimma University and the authors did not received financial support for the publication of the article.

**Competing interests:** The authors have declared that no competing interests exist.

effects, screening patients for physical illness, and psycho-education on substance use should be given top priority.

## Introduction

Drug-induced movement disorders (DIMD) are neurological motor disturbances that most frequently associated with drugs that block dopamine (D2) receptors [1]. Drug-induced movement disorders can be caused by different kinds of agents, and almost all kinds of movement disorders can happen as a result of a medication side effect. Responsible psychotropic drugs include antipsychotics, antidepressants, mood stabilizers, and anti-convulsant [2]. Psychotropic medication-induced movement disorders (PMIMD) often expose patients to stigma and can impair a patient's ability to complete activities of daily living [3]. Side effects are common, and the severity ranges from tremor to life-saving syndromes [4]. Drug-induced parkinsonism (DIP) and akathisia (DIA) usually develop over days to weeks or months following the ingestion of responsible drugs. DIP is characterized by akinetic rigidity, bradykinesia, and postural instability. The classic feature of DIP is symmetrically distributed syndrome and high-amplitude chin and jaw tremor [5]. DIA is defined as a subjective complaint of restlessness, often accompanied by observed excessive movements like fidgety movements of the legs, rocking from foot to foot, pacing, inability to sit or stand still that develops within a few weeks of starting or raising the dosage of a medication (such as a narcoleptic) [6].

The magnitude of DIMD according to the Diagnostic and Statistical Manual (DSM- IV) criteria was 61.6% [7]. Studies conducted in Europe reported a prevalence which ranges from 22% to 31.7% [8, 9]. A cross-sectional study done in Poland reported the magnitude of DIP and DIA to be 22.9% and 24.5% respectively, [10]. The magnitude was even much higher in Africa in which case prevalence of DIP ranges from 18.8% to 66.7% [11]. According to a prospective study done in a Nigerian hospital, the prevalence rate of DIP was 27.2% and 14.6% for DIA [12], and a study done in Ethiopia reported a higher magnitude (DIP 46.4% and DIA 28.6%) [13].

The magnitude of DIP and DIA varies with the duration of treatment, type of medication, sex, age, and dosage of medication. In 50–75% of cases and 90% of cases develop DIP after a month of treatment and three-month treatment, respectively, [14]. Similarly, the incidence is reported to increase with age, and the highest incidence is reported in the age range of 60–80 years [15]. Another study reported that older people were about five times more likely to develop DIP [16]. Regarding sex as a risk factor, studies report conflicting evidence. According to a prospective study done in Nigeria, the risk/ratio of developing AIPD with male to female ratio of 2:1 [12]. Similar studies reported a positive association between female sex and DIP [14, 17, 18]; however, other studies did not find a difference in gender and DIP [13, 19].

Overall, antipsychotic-associated movement disorders have several impacts on patients. It can exert a negative impact on patients' quality of life, medication adherence, and social well-being of the patient [20]. The intolerability of these side effects often leads to a relapse of psychiatric symptoms following drug non-adherence, which in turn results in joblessness and poor progression of illness and increased suicidal risk [21]. Determining the magnitude and identifying factors associated with a drug-induced movement disorder has paramount importance in preventing negative physical, emotional, and social consequences related to psychotropic medication-induced movement disorder. Despite the magnificent effect of the problem, there are limited studies in this area. Thus, this study aimed to assess the prevalence and factors

associated with drug-induced Parkinsonism and akathisia in a patient with mental illness taking psychotropic medication at Jimma Medical Center (JMC), psychiatry follow-up clinic, Southwest, Ethiopia.

## Materials and methods

### Study design and setting

A hospital-based cross-sectional study design was employed at JMC from April to June 2019. JMC is one of the specialized medical centers and is located 352 km southwest of Addis Ababa, the capital of Ethiopia. The center renders various medical services at the outpatient department (OPD) and in-patient department (IPD) setup for approximately 15 million populations in Southwest Ethiopia. The psychiatric clinic of JMC was established in 1996. Currently, there are more than one thousand psychiatric patients attending follow-up treatments, and about fifty patients visit psychiatric clinics daily.

### Population

All patients diagnosed with mental illness and attending follow-up treatment with psychotropic medication at the outpatient department of the JMC psychiatric clinic were a source population. Study participants were randomly selected from mental ill patients who were taking psychotropic medications. Accordingly, Four hundred twenty adult persons (age 18 and above) who were diagnosed with mental illness according to the diagnostic criteria of the Diagnostic Statistical Manual (DSM-IV or DSM-5) were included in the study. Mentally ill patients who were unable to give the required information, who had a history of primary movement disorders, and who had medically established brain lesions were excluded from the study.

### Sample size and sampling techniques

The minimum number of the sample size required for this study was determined by using the formula to estimate a single population proportion. The minimum sample size determination formula used is: $(n) = \frac{\left(\frac{Z\alpha}{2}\right)^2 p(1-p)}{d^2}$, where $n$ denotes the minimum sample size, Z $\alpha$/2 is the reliability coefficient of the standard error at a 5% level of significance = 1.96, p is the proportion of mentally ill patients who were taking psychotropic medication and developed drug-induced parkinsonism = 46.4% [13], and 10% non-response rate used. Hence, the minimum sample size obtained is 420. Accordingly, study participants were recruited using a systematic random sampling technique. A sampling fraction was determined by dividing the average number of patients attending follow-up treatment at the clinic by the calculated minimum sample size required. A sample fraction of two was used, and every two study participants were approached, and the first study participant included in the study was determined by the lottery method. Patient medical record numbers were used as a code to avoid repeated involvement of a case in the study.

### Data collection procedure and instrument

Data were collected through face-to-face interviews using semi-structured and pre-tested questionnaires. The questionnaire was primarily prepared in English and translated into Afaan Oromo and Amharic. Re-translation of the questionnaire back to English was done by another person who was fluent in English language and new for the original version. The extrapyramidal symptom rating scale (ESRS) was used to assess the presence of drug-induced movement disorders [22]. The Extra Pyramidal Symptom Rating Scale was developed to assess four types

of drug-induced movement disorders: drug-induced akathisia, drug-induced parkinsonism, drug-induced dystonia, and drug-induced tardive dyskinesia. The tool has four subscales and four Clinical Global Impression Scales (CGI-S).

Patients subjective reports were obtained through face-to-face interviews and it was scored on a 4-point scale (0 = absent; 1 = mild; 2 = moderate; 4 = severe). The data collectors considered verbal reports with frequency and duration of symptoms, several days the symptoms present, and the intensity of symptoms over the last 7 days. For Parkinsonism and akathisia examination, the presence and/or absence of tremor, bradykinesia, gait and posture, postural stability, rigidity, and expressive automatic movement and akathisia were evaluated by trained data collectors. Tremor and rigidity (items 1 and 5) were scored on a 7-point item scale (0 = none and 6 = severe). The tremor was rated for each part of the body (right and left upper and lower limbs, head, tongue, jaw, and lips) separately, and for rigidity, the rating was done for the right and left upper and lower limbs separately. The total score ranges from 0–96 (16 items), and two factors: hypokinesia (0–42) and hyperkinesia (0–49). The score for akathisia (0–6) is based on the sum score of subjective akathisia (item 6) and objective finding (item 7).

Compared to the Simpson-Angus scale, which does not focus on rating tremors on the part of the body affected, and the akathisia item of Simpson-Angus, which has been added in some version of the scale, focuses on objective evidence only, ESRS has well-established validity [23]. The tool demonstrated high inter-rater reliability (range of mean item correlation coefficients 0.80 to 0.97) [22].

CGIS was developed to provide a brief assessment of the clinician's view of the patient's functioning and severity of symptoms [24]. The tool provides a clinically determined measure of symptom severity, behavior, and the impact of the symptoms on the patient's ability to function. The tool comprises one-item measures evaluating the severity of psychopathology and allows the subjective evaluation of symptoms by applying an 8 point rating (0 = absent; 1 = borderline; 2 = very mild; 3 = mild; 4 = moderate; 5 = moderate to severe; 6 = marked; 7 = severe; 8 = extremely severe). Substance use was assessed using the World Health Organization Alcohol, Smoking, and Substance Involvement Screening test (WHO ASSIST) version 3.1. It has 8-items and is culturally neutral and usable across a variety of cultures to assess the use of the psychoactive substance. The total item score ranges from 0 to 31 for tobacco and 0–39 for alcohol and khat (Catha edulis). Total scores for khat, tobacco, and cannabis were categorized as low (score 0–3), moderate (score 4–26), and high (score $\geq 27$) [25]. Furthermore, the study participants' socio-demographic such as age, sex, ethnicity, religion, marital status, educational status, occupational status, and income assessed using a structured questionnaire. Medication-related factors like the type of medication, dosage, frequency, and duration of medication; and clinical-related factors such as type of diagnosis, presence of comorbid psychiatric and/or physical illness (specifically diabetes mellitus and hypertension) were assessed from the patient's medical record review. For data collection, four Bachelor of Science (BSc) Psychiatric nurses were recruited to conduct face-to-face interviews. Two days of training on data collection tools, patients' objective evaluation, and ethical issues were given to data collectors and supervisors. The pretest was conducted on 20 patients attending follow-up treatment at *Shenan Gibe* General Hospital, located 5 kilometers (km) from JMC to Southwest.

## Data processing and analysis

The data were edited, cleaned, coded, and entered into the Epi-data 3.1 version and analyzed using Statistical Package for Social Sciences (SPSS) version 22. Binary logistic regression was used for comparison of the subjects with and without DIP and DIA. Factors which found to have a *p*-value's of less than 0.25 were a candidate for multivariable logistic regression analysis,

and a $p$-value $\leq 0.05$ was used to declare a statistically significant association. The strength of the association was presented by odds ratio with 95% confidence interval (CI). The Hosmer-Lemeshow goodness-of-fit test was used to check model fitness. Multi-collinearity was checked using a variance inflation factor (VIF). Finally, results are presented in the form of figure and tables using frequency and summary statistics such as mean and percentage.

### Ethical consideration

Ethical clearance was obtained from the Research Ethical Review Board (IRB) of Jimma University, Institute of Health, and the research was performed according to the Declaration of Helsinki. Written informed consent was obtained from each study participant. Study participants were informed of the right to refuse or discontinue participation at any time they wanted, and the chance was given to ask any thing about the study. The filled questionnaires were kept securely locked. Participants who were found to have movement disorders were linked to psychiatrists for consultation.

## Results

### Socio-demographic description of participants

The response rate of the study was 97.3% ($n$ = 410). The mean age of the respondents was 33.3 years (SD ± 8.55). About two-thirds 263 (64.1%) of the study participants were male. More than half of the study participants were single and resided in urban areas (**Table 1**).

### Clinical and medication-related characteristics of the participants

Most of the study participants 223 (54.4%) had a diagnosis of schizophrenia. Three-fifths of the study participants 246 (60.0%) had received treatment for 1–5 years with a mean length of 4.8 years (SD = 3.9). Regarding the type of psychotropic medications, 240 (58.5%) patients were receiving typical antipsychotic medications, and 187 (45.6%) patients received anticholinergic drugs in combination with antipsychotic medications. The mean daily chlorpromazine equivalent dose was 425mg (SD = 245). Among patients who had drug-induced parkinsonism and akathisia, 32 (54.2%) and 27 (52.9%) used typical antipsychotics, respectively (**Table 2**).

### Substance use-risk

Among the study participants, 164 (39.9%) had used substances at least once in their lifetime, and 99 (24.1%) used substances in the last three months before the study period. One-fifth 70 (17.1%) and one-tenth 32 (7.8%) of the study participants met moderate and severe-risk levels of khat use, respectively. Regarding tobacco, 118 (28.9%) and 100 (24.4%) study participants met moderate and severe-risk levels of tobacco use, respectively, (**Table 2**).

### Magnitude of drug-induced parkinsonism and akathisia

The prevalence of DIP was 14.4% (95% CI: 11.0 to 18.0) and it was 12.4% (95% CI: 9.3 to 15.4) for DIA. The most commonly reported subjective items were restlessness 52 (12.7%) and tremor of the hand 70 (17.1%). On examination, the most prevalent signs of Parkinsonism were tremor 80 (19.5%) and gait and posture disturbance 56 (13.7%), (**Fig 1**).

### Multivariate regression analysis

During bi-variable logistic regression analysis of drug-induced parkinsonism in relation to each explanatory variable: sex, age, duration of treatment, type of antipsychotics, dose of

**Table 1. Socio-demographic characteristic distributions of mental ill patients attending follow-up treatment at JMC, 2019.**

| Variables | Categories | Frequency(N = 410) | Percent (%) |
|---|---|---|---|
| Age(year) | 15–29 years | 145 | 35.4 |
| | 30–44 years | 201 | 49.0 |
| | > 45years | 64 | 15.6 |
| Sex | Male | 263 | 64.1 |
| | Female | 147 | 35.9 |
| Marital status | Married | 168 | 41 |
| | Single | 213 | 52 |
| | Divorced | 28 | 6.8 |
| | Widowed | 1 | 0.2 |
| Religion | Muslim | 222 | 54.1 |
| | Orthodox | 120 | 29.3 |
| | Protestant | 67 | 16.3 |
| | Others | 1 | 0.2 |
| Ethnicity | Oromo | 237 | 57.8 |
| | Amhara | 107 | 26.1 |
| | Tigre | 12 | 2.9 |
| | Dawuro | 39 | 9.5 |
| | Others | 15 | 3.6 |
| Residence | Rural | 204 | 49.8 |
| | Urban | 206 | 50.2 |
| Educational status | No formal education | 90 | 22.0 |
| | 1–4 grade | 126 | 30.7 |
| | 5–8 grade | 109 | 26.6 |
| | 9–12 grade | 52 | 12.7 |
| | College and above | 33 | 8.0 |
| Occupational status | With job | 192 | 46.8 |
| | Jobless | 218 | 53.2 |

Other religion = Catholic and Wakefata; other ethnicity = Silte, Wolayta, and Yeme.

antipsychotics, physical illness, and medication for movement disorders were the variables that fulfilled the minimum requirement (in this study, 0.25 level of significance) and entered into multivariate logistic regression analysis. Results of multivariate logistic regression analysis showed that drug-induced Parkinsonism was significantly associated with sex, age, type of antipsychotics, physical illness, and anti-cholinergic medication use. The odds of developing drug-induced parkinsonism among female patients were 2.3 times higher than among male patients. Patients aged ≥ 45 years were 4.3 times more likely to develop antipsychotic-induced parkinsonism compared to those patients aged 15–29 years. Patients who had physical illness were four times more likely to develop drug-induced parkinsonism compared to those who had no physical illness. Participants taking anti-cholinergic medications were 88% more likely to develop drug-induced parkinsonism compared to their counterparts. Those patients who received typical antipsychotics were 2.92 times and those taking both typical and atypical were 1.4 times more likely to develop DIP as compared to those patients who were taking atypical antipsychotic medications (**Table 3**).

Similarly, in the final model, being female, chlorpromazine equivalent doses of 200–600 mg, adjuvant use of sodium valproate, and severe khat use risk level were significantly associated with drug-induced akathisia. Females were approximately four times more likely to

**Table 2. Clinical, medication-related, and substance use risk levels of patients with mental illness attending follow-up treatment at JMC, 2019.**

| Characteristics | Frequency | Percent |
|---|---|---|
| **Diagnosis of patients** | | |
| Schizophrenia | 223 | 54.4 |
| Major depressive disorder | 99 | 24.1 |
| Bipolar I disorder | 74 | 18.0 |
| Other schizophrenia spectrum disorder | 14 | 3.4 |
| **Duration of treatment** | | |
| < 1 years | 37 | 9.0 |
| 1–5 years | 246 | 60.2 |
| 5 years | 127 | 31.0 |
| **Physical illness** | | |
| Yes | 38 | 9.3 |
| No | 372 | 90.7 |
| **Types of antipsychotics** | | |
| Typical | 240 | 58.5 |
| Atypical | 95 | 23.2 |
| Both | 75 | 18.3 |
| **Anticholinergic medications use** | | |
| Yes | 187 | 45.6 |
| No | 223 | 54.4 |
| **Khat use risk level** | | |
| No/mild | 308 | 75.1 |
| Moderate | 70 | 17.1 |
| High | 32 | 7.8 |
| **Tobacco use risk level** | | |
| No/mild | 192 | 46.7 |
| Moderate | 118 | 28.9 |
| High | 100 | 24.4 |

Physical illness = diabetic mellitus and/or hypertension.

develop DIA as compared to male patients. Participants who were treated with chlorpromazine equivalent doses of 200–600 mg were 1.36 times more likely to develop drug-induced akathisia than those treated with less than 200 mg doses. Patients receiving sodium valproate in combination with antipsychotics were 1.88 times more likely to develop akathisia than those receiving only antipsychotics. Furthermore, severe risk level of *khat* use disorder was also significantly associated with drug-induced akathisia (**Table 4**).

## Discussion

### Prevalence of drug induce parkinsonism and akathisia

The prevalence of drug-induced parkinsonism and akathisia in this study found 14.4% (95% CI: 11.0 to 18.0) and 12.4% (95% CI: 9.3 to 15.4), respectively. The magnitude of DIP was lower than studies done in California 30% [26], the Netherlands 56.2% [27], the United Kingdom 26% [28], Poland 22.99% [10], Estonia 23.2% [29], Nigeria 27.2% [12], and Ethiopia 46.4% [13]. This could be a variation in the study population, the difference in assessment tools, and variation in the type of psychotropic medication. In a study conducted in Estonia,

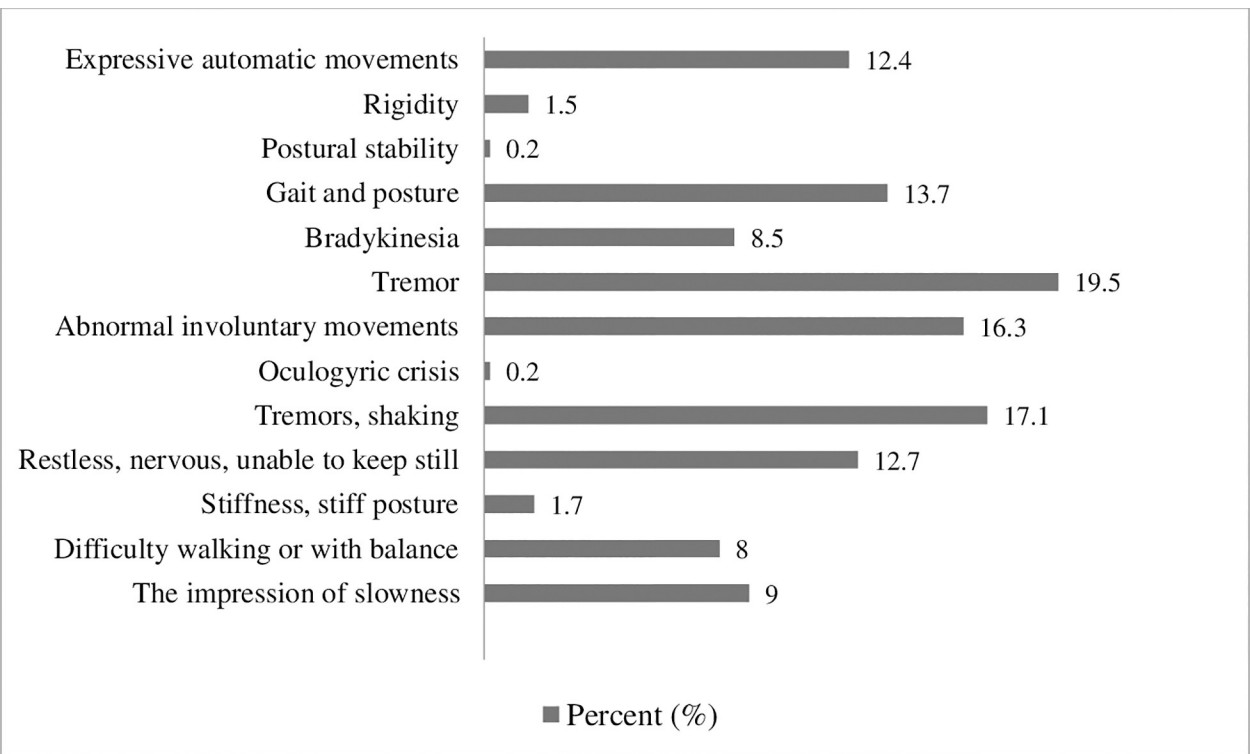

**Fig 1. Percentage distribution of drug induced signs and symptoms among mentally ill patients attending follow-up treatment at JMC Psychiatry clinic, 2019.**

Nigeria, Ethiopia, and Poland, the study was conducted only on Schizophrenia patients who were taking antipsychotic medication. DIMD is associated with a more vulnerable dopamine system [30] and the motor symptoms mostly occur after the introduction of dopamine receptor-blocking antipsychotics. Thus, patients with schizophrenia are at increased risk of developing extra-pyramidal symptoms specifically, parkinsonism [27]. Furthermore, the relatively lower magnitude of DIP in the current study might be due to the late introduction of second-generation antipsychotics. In this study most of the study participants used first-generation antipsychotics, which carry a higher risk of side effects. However, the contrary finding is reported in France [31] and Saudi Arabia [32]. Regarding DIA, the magnitude found in this study is higher than the study done in the Netherlands 4.6% [27], California 7% [26], the United Kingdom 1.3% [28], and France 1.2% [31]. The discrepancy may be due to the difference in tools used and the relatively high daily dosage and type of psychotropic medication used.

## Factors associated with DIP and DIA

During the analysis of drug-induced parkinsonism, female patients were significantly associated with DIP (AOR = 2.3, $p$ = 0.035) and DIA (AOR = 3.9, $p$ = 0.003) compared to male patients. This might be due to striatal dopaminergic and cholinergic systems being under regulatory control by estrogen. In females, the balance between these two neurotransmitter systems in the striatum may be shifted towards higher cholinergic activity, a condition that favors the development of movement disorders in females. This was supported by other studies done by other studies [12, 32]. In the case of DIP, patients aged range between 30 to 44 years were 1.51 times and ≥ 45 years 2.9 times (AOR = 2.9, $p$ = 0.011) more likely to develop DIP as compared

**Table 3. Factors associated with drug-induced parkinsonism among patients taking antipsychotics and attending follow-up treatment at JMC, 2019.**

| Explanatory variables | Drug-induced parkinsonism | | COR (95% CI) | AOR (95% CI) | p-value |
|---|---|---|---|---|---|
| | Yes | No | | | |
| **Age** | | | | | |
| 15–29 | 15 | 130 | 1 | 1 | |
| 30–44 | 22 | 179 | 1.52 (0.87, 2.65) | 1.51 (1.17, 3.12) | 0.004* |
| ≥ 45 | 22 | 42 | 2.46 (1.34, 4.52) | 2.9 (2.5, 8.4) | 0.011* |
| **Sex** | | | | | |
| Male | 38 | 225 | 1 | 1 | |
| Female | 21 | 126 | 3.42 (2.2, 5.4) | 2.3 (1.06, 5.07) | 0.035* |
| **Duration of treatment** | | | | | |
| < 1 year | 10 | 27 | 1 | 1 | |
| 1–5 years | 23 | 222 | 1.52 (0.87, 2.65) | 1.55 (0.17, 3.12) | 0.482 |
| ≥ 5years | 26 | 102 | 2.41 (1.52, 5.57) | 2.8 (0.78, 4.61) | 0.079 |
| **Type of antipsychotics** | | | | | |
| Typical | 32 | 208 | 3.49 (1.96, 4.2) | 2.92 (1.34, 4.71) | 0.003* |
| Atypical | 18 | 77 | 1 | 1 | |
| Both | 11 | 64 | 1.15 (1.12, 2.48) | 1.42 (1.24, 3.14) | 0.013* |
| **Recent Anti-cholinergic use** | | | | | |
| Yes | 48 | 139 | 0.15 (0.08, 0.29) | 0.12 (0.05, 0.29) | 0.004* |
| No | 11 | 212 | 1 | 1 | |
| **Co-morbid physical illness** | | | | | |
| Yes | 24 | 14 | 2.12 (1.28, 3.50) | 4 (2.2, 8.2) | 0.002* |
| No | 35 | 337 | 1 | 1 | |
| **Type of psychiatry diagnosis** | | | | | |
| Schizophrenia | 15 | 208 | 1 | 1 | |
| Major depression | 7 | 92 | 2.97 (1.3,2.46) | 4.76 (2.65,7.22) | 0.014* |
| Bipolar I disorder | 29 | 45 | 3.11 (1.05,7.2) | 2.3 (0.071, 3.4) | 0.063 |
| Other schizophrenia spectrum disorders | 2 | 12 | 1.2 (0.30, 7.16) | 1.7 (0.23, 6.34) | 0.925 |

1 = reference group; co-morbid physical illness = diabetic mellitus and/or hypertension

*$p < 0.05$.

to those patients aged range 15–29 years. This was supported by a study done in India and the Netherlands [10, 32]. In normal aging, evidence indicates that dopamine displays a special vulnerability. It is indicated that there is a decrease of 4.7% dopamine neurons per decade in the nigrostriatal pathway in normal human brain [33], which further increases the risk of DIP as age increases. A study done by *Margaret et al.* [34] reported that conventional antipsychotic-induced extrapyramidal side effects increased three to four times as age increased [35].

Patients taking typical antipsychotics were about three times (AOR = 2.92, $p$ = 0.003) more likely to develop drug-induced parkinsonism. This was supported by a study done in Saudi Arabia [32] and Nigeria [12]. Binding of first-generation antipsychotics (FGA) on dopamine 2 (D2) receptors is related to higher acute extrapyramidal symptoms (EPS) [36] than second-generation antipsychotics (SGA), which loosely binds and dissociates rapidly to D2 receptors [37]. However, another study did not find a statistically significant difference between patients taking FGA and SGA in terms of emergent parkinsonism, akathisia, or tardive dyskinesia at either follow-up point [38]. Those who were treated with chlorpromazine equivalent doses of 200 to 600 mg were 1.36 times (AOR = 1.36, $p$ = 0.008) more likely to develop drug-induced akathisia than those treated with less than 200 mg doses. The same finding was reported in

**Table 4. Factors associated with drug-induced akathisia among patients taking antipsychotics and attending follow-up treatment at JMC, 2019.**

| Explanatory variables | Drug-induced akathisia | | COR (95% CI) | AOR (95% CI) | P-value |
|---|---|---|---|---|---|
| | Yes | No | | | |
| **Sex** | | | | | |
| Male | 25 | 237 | 1 | 1 | |
| Female | 25 | 122 | 1.9 (1.53, 3.37) | 3.9 (2.6, 9.5) | 0.003* |
| **Type of psychiatry diagnosis** | | | | | |
| Schizophrenia | 15 | 208 | 1 | 1 | |
| Major depression | 7 | 92 | 2.97 (1.3,2.46) | 4.76 (2.65,7.22) | 0.014* |
| Bipolar I disorder | 29 | 45 | 3.11 (1.05,7.2) | 2.3 (0.071, 3.4) | 0.063 |
| Other schizophrenia spectrum disorders | 2 | 12 | 1.2 (0.30, 7.16) | 1.7 (0.23, 6.34) | 0.925 |
| **Chlorpromazine equivalent dose** | | | | | |
| < 200mg | 6 | 137 | 1 | 1 | |
| 200-600mg | 45 | 218 | 1.19(0.08, 1.48) | 1.36(1.17,3.7) | 0.008* |
| >600mg | 1 | 3 | 2.2(0.6, 4.9) | 3.19(0.13, 9.06) | |
| **Other psychotropic medications than antipsychotics** | | | | | |
| Sodium valproate | 15 | 212 | 2.14(0.4,11.4) | 1.88(1.3, 11.8) | 0.023* |
| Amitriptyline | 7 | | 3.06(1.1,8.4) | 1.26(0.09, 2.88) | |
| Sodium and Carbamazepine | 10 | 14 | 7.3(3.28,12.5) | 1.9(1.3, 4.8) | 0.010* |
| Not taking other medications | 17 | 337 | 1 | 1 | |
| **Recent Anti-cholinergic use** | | | | | |
| Yes | 41 | 146 | 1.7 (0.08, 3.3) | 2.1(1.09, 4.9)* | 0.009* |
| No | 10 | 213 | 1 | 1 | |
| **Khat use risk level** | | | | | |
| No/mild | 24 | 284 | 1 | 1 | |
| Moderate | 17 | 53 | 3.49 (1.89, 6.43) | 1.89 (0.7, 3.78) | 0.312 |
| High | 17 | 15 | 4.98 (2.7, 8.79) | 3.1 (2.11, 6.67) | 0.03* |

1 = reference group

*$p < 0.05$.

Ethiopia [13]. Patients taking sodium valproate were 1.88 times more likely to develop akathisia than those taking antipsychotics only; however study done by *Weng J et al.* did not find out any statistically significant difference in extra-pyramidal symptoms between the two groups of patients [39].

The association between substance use risk level and DIP/DIA was assessed in this study. Accordingly, it is only severe *khat (Catha edulis)* use risk level and DIA, which is significantly associated. Participants with severe khat use risk level were found to be three times (AOR = 3.1, $p$ = 0.03) more likely to develop DIA than those with no/mild risk level. A study done in Ethiopia reported a positive association between khat use and DIA [13]. This association might be due to patients who were with severe risk levels may develop more withdrawal symptoms that mimic the symptoms of akathisia. Furthermore, khat is assumed to have similar actions as amphetamine, which increases striatal dopamine release in the acute phase. Furthermore, in this study, patients diagnosed with major depressive disorder were about five times (AOR = 4.76, $p$ = .014) more likely to develop DIP compared to patients with schizophrenia. This might be due to most patients with major depressive disorder in the current study were taking typical antipsychotics together with antidepressants, which carries a higher risk of DIP.

However, no statistically significant association was identified between DIA and the type of psychiatric diagnosis. Patients with comorbid physical illness were four times (AOR = 4.01, $p$ = 0.002) more likely to develop drug-induced Parkinsonism, and use of anti-cholinergic medication was 88% (AOR = 0.12, $p$ = 0.004) protective for DIP as compared to those who did not take anti-cholinergic medications.

## Conclusions

One every seven patients developed DIP and DIA. Sex, age, and comorbid physical illnesses were found to be associated with drug-induced Parkinsonism. The dose of mediation, khat use, major depressive disorder, and sodium valproate were positively associated with Akathisia. Physical illness was associated with Parkinsonism. Thus, careful assessment of patients for DIP/DIA, selection of drugs with minimal side effects, screening patients for physical illness, and psycho-education on substance use should be given top priority.

## Acknowledgments

The authors would like to acknowledge the study participants, data collectors, and supervisors who spent their valuable time for the good outcome of this research.

## Author Contributions

**Conceptualization:** Assefa Kumsa.

**Data curation:** Assefa Kumsa, Liyew Agenagnew, Shimelis Girma.

**Formal analysis:** Assefa Kumsa, Liyew Agenagnew, Beza Alemu, Shimelis Girma.

**Funding acquisition:** Assefa Kumsa.

**Investigation:** Assefa Kumsa, Liyew Agenagnew, Beza Alemu, Shimelis Girma.

**Methodology:** Assefa Kumsa, Beza Alemu, Shimelis Girma.

**Project administration:** Assefa Kumsa.

**Resources:** Assefa Kumsa.

**Software:** Assefa Kumsa, Liyew Agenagnew, Beza Alemu, Shimelis Girma.

**Supervision:** Assefa Kumsa.

**Validation:** Liyew Agenagnew, Beza Alemu, Shimelis Girma.

**Writing – original draft:** Shimelis Girma.

**Writing – review & editing:** Assefa Kumsa, Liyew Agenagnew, Beza Alemu, Shimelis Girma.

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
