## [Decision Letter · Decision Letter 0]

26 May 2020

PONE-D-20-06567

Psychotropic Medications Induced Parkinsonism and Akathisia in People Attending Follow-up Treatment at Jimma Medical Center Psychiatry Clinic

PLOS ONE

Dear Dr. Girma,

Thank you for submitting your manuscript to PLOS ONE. After careful consideration, we feel that it has merit but does not fully meet PLOS ONE’s publication criteria as it currently stands. Therefore, we invite you to submit a revised version of the manuscript that addresses the points raised during the review process.

Please, be aware that submitting a reviewed paper does not guarantee acceptance.

We look forward to receiving your revised manuscript.

Kind regards,

Vincenzo De Luca

Academic Editor

PLOS ONE

Journal Requirements:

3. Your ethics statement must appear in the Methods section of your manuscript. If your ethics statement is written in any section besides the Methods, please move it to the Methods section and delete it from any other section. Please also ensure that your ethics statement is included in your manuscript, as the ethics section of your online submission will not be published alongside your manuscript.

"Financial expense for data collection was covered by Jimma University and the authors did not

receive any financial support for the publication of the article."

"No authors received no specific funding for this work."

Reviewers' comments:

Reviewer's Responses to Questions

**Comments to the Author**

1. Is the manuscript technically sound, and do the data support the conclusions?

Reviewer #1: Yes

2. Has the statistical analysis been performed appropriately and rigorously? 

Reviewer #1: No

3. Have the authors made all data underlying the findings in their manuscript fully available?

Reviewer #1: Yes

4. Is the manuscript presented in an intelligible fashion and written in standard English?

Reviewer #1: Yes

5. Review Comments to the Author

Reviewer #1: This is an interesting article discussing an understudied phenomenon in a specific population. I believe the article will be suitable for publication in Plos One once the authors provide some clarifications. A careful revision of the manuscript for typos and unclear wording is also warranted.

1. My biggest concern is related to why the authors did not include statistical analyses on the correlation of DIP/DIA with specific disorders and/or severity of symptoms of such disorders. I believe this is an obvious point of discussion, and more importantly a clinical indication. It would greatly strengthen the analysis and the discussion which feels somewhat limited to epidemiological findings in the sample (e.g. 55% were schizophrenia, etc.). Some of that data seems to be present in the tables, however it is not made clear enough.

2. Another interrogation is the concept of use of substances here, which is not properly separated from the DSM definition of substance use disorder (SUD). The sample suggests that some used substances in the past and others were constant users. However, given the sample, I think SUDs should be addressed.

3. What are the columns yes/no in the tables?

4. Some precisions on the “physical illness” are needed. Are we including neurological lesions, etc.? This is too vague.

5. I don’t understand the sentence describing the 4.7% decrease of dopaminergic neurons with aging. This should be made more clear.

6. PLOS authors have the option to publish the peer review history of their article (what does this mean?). If published, this will include your full peer review and any attached files.

Reviewer #1: No

---

## [Author Response · Author response to Decision Letter 0]

3 Jun 2020

Dear Editor, 

We would like to say thank you for assigning reviewer and returning the crucial comments and recommendations. We found the points raised by the reviewer are very interesting to make this article scientifically sound. We addressed all the comments of the editor and reviewer, and we wrote clarifications here under for mutual understanding. 

Editorial comments: the following points are addressed.

1. Ethical statement: As per the editor comment and journal requirement, ethical statement is included in the method section (P #10, line 180- 

 187) of the revised manuscript. Accordingly, ethical statement is deleted from the other section. 

2. Funding section: Thank you for your critical observation and apology for inconsistency in this regard. The funding information that appears in 

 the acknowledgment section is corrected and it is not stated in any part of the revised manuscript. I kindly request to update the funding 

 statement in the online submission form as “Financial expense for data collection was covered by Jimma University. The authors have not 

 received financial support for the publication of the article.”

3. Language usage, spelling, grammar, and journal formatting style: As per the recommendation of the reviewer, extensive language revision is 

 made in the current document. In the current revised document through formatting of the entire manuscript is done. A copy of document 

 showing changes (language usage, spelling, grammar, and formatting style) is indicated by track changes and attached as ‘Revised 

 Manuscript with Track Changes.

Reviewers’ comments and authors response 

1. ‘Why authors did not include statistical analysis on the correlation of DIP/DIA with specific disorders and/or severity of symptoms of such disorders…..’

Response: Thank you for the concern and we found this comment very important in lens of clinical implication. The researchers included type of psychiatry diagnosis in the final model of DIA, as there was independent association between the two (P #20, table 3). The binary regression between type of psychiatry diagnosis and DIP, found out no statistical significant association between the two. Thus the explanatory variable was removed during multivariable regression analysis and not presented in the table. The discussion point is supplemented in line with the finding and it is added in the last paragraph of discussion section in the revised manuscript (P #15, line 294-97 )

2. ‘The concept of use of substance is not properly separated from the DSM definition of substance use disorder and thus the issue of SUD should be addressed’ 

Response: Thank you for the comment; we appreciate the critical observation of the reviewer. The researchers used WHO ASSIST screening tool which allows identification of risk level of substance use. We did not engage structured clinical interview with DSM to identify substance use disorder. Thus, the current manuscript is revised to address the risk level rather than focusing simply on substance use; which is the point of concern of the reviewer. In line with the comment of the reviewer we integrate operation definition of each risk level in method section (P # 9, line 157-59), result section is modified in P # 11, line 201-06) and the discussion is revised in P # 14-15, line 286-93). We hope that this increase the scientific sound of the result. 

3. ‘What are the columns yes/no in the tables?’ 

Response: Thank you once again; we thought that the vague formatting of the tables leads to this question. We formatted the final table (P #19-20) to clearly indicated for what yes/no stands 

4. ‘Some precision on the “physical illness” are needed. Are we including neurological lesions, etc’ 

Response: Sure, it is nice point. In this study we only focused on presence of chronic non-communicable disease specifically diabetic and/or hypertension. This is clarified in data collection procedure and instrument sub-section of method part of the document (P #9, line 164). Furthermore, patient who had any primary movement disorder and brain lesion was excluded from the study and it is stated in population sub-section of methods and materials (P #6, line 105-06)

5. ‘I don’t understand the sentence describing the 4.7% decrease of dopaminergic neuron with aging. This should be made more clear.’ 

Response: Sorry for the clarity problem of the statement. The statement is corrected for easy understandability (PP # 14, line 268-70). 

Thank you in advance! 

Shimelis Girma 

Corresponding author

---

## [Decision Letter · Decision Letter 1]

15 Jun 2020

Psychotropic medications induced parkinsonism and akathisia in people attending follow-up treatment at Jimma Medical Center, Psychiatry Clinic

PONE-D-20-06567R1

Dear Dr. Girma,

We’re pleased to inform you that your manuscript has been judged scientifically suitable for publication and will be formally accepted for publication once it meets all outstanding technical requirements.

Kind regards,

Vincenzo De Luca

Academic Editor

PLOS ONE

Additional Editor Comments (optional):

Reviewers' comments:

Reviewer's Responses to Questions

**Comments to the Author**

1. If the authors have adequately addressed your comments raised in a previous round of review and you feel that this manuscript is now acceptable for publication, you may indicate that here to bypass the “Comments to the Author” section, enter your conflict of interest statement in the “Confidential to Editor” section, and submit your "Accept" recommendation.

Reviewer #1: All comments have been addressed

2. Is the manuscript technically sound, and do the data support the conclusions?

Reviewer #1: Yes

3. Has the statistical analysis been performed appropriately and rigorously? 

Reviewer #1: Yes

4. Have the authors made all data underlying the findings in their manuscript fully available?

Reviewer #1: No

5. Is the manuscript presented in an intelligible fashion and written in standard English?

Reviewer #1: Yes

6. Review Comments to the Author

Reviewer #1: The authors have answered all my comments. However, I suggest the authors carefully review the manuscript before submission as some sentences are still hard to understand.

7. PLOS authors have the option to publish the peer review history of their article (what does this mean?). If published, this will include your full peer review and any attached files.

Reviewer #1: No

---

## [Editor Report · Acceptance letter]

23 Jun 2020

PONE-D-20-06567R1 

Psychotropic medications induced parkinsonism and akathisia in people attending follow-up treatment at Jimma Medical Center, Psychiatry Clinic 

Dear Dr. Girma:

I'm pleased to inform you that your manuscript has been deemed suitable for publication in PLOS ONE. Congratulations! Your manuscript is now with our production department. 

Kind regards, 

on behalf of

Dr. Vincenzo De Luca 

Academic Editor

PLOS ONE